# The Fate and Intermediary Metabolism of Soyasapogenol in the Rat

**DOI:** 10.3390/molecules28010284

**Published:** 2022-12-29

**Authors:** Chenghui Pan, Yonggang Yan, Dayun Zhao

**Affiliations:** 1Department of Food Science and Technology, School of Agriculture and Biology, Shanghai Jiao Tong University, 800 Dongchuan Road, Shanghai 200240, China; 2Bor S. Luh Food Safety Research Center, Shanghai Jiao Tong University, 800 Dongchuan Road, Shanghai 200240, China

**Keywords:** soyasapogenol, metabolite profiling, bioavailability, LC/MS–QTOF

## Abstract

Research suggests that soyasaponins are poorly absorbed in the GI tract and that soyasaponin aglycones or soyasapogenols are absorbed faster and in greater amounts than the corresponding soyasaponins. Therefore, it is important to understand the bioavailability of these compounds for the potential development of functional foods containing their components. In this paper, to investigate the metabolic characteristics of soyasapogenols A and B, the pharmacokinetic parameters in rats were determined via oral and intravenous administration. The liver metabolites of soyasapogenols were identified using UPLC–/Q-TOF–MS/MS, and their metabolic pathways were also speculated. The results show that, after oral administration, there was a bimodal phenomenon in the absorption process. T_max_ was about 2 h, and soyasapogenol was completely metabolized 24 h later. The bioavailability of soyasapogenol was superior, reaching more than 60%. There were sixteen metabolites of soyasapogenol A and fifteen metabolites of soyasapogenol B detected in rat bile. Both phase I and II metabolic transformation types of soyasapogenols, including oxidation, dehydrogenation, hydrolysis, dehydration, deoxidization, phosphorylation, sulfation, glucoaldehyde acidification, and conjugation with cysteine, were identified. In addition, soyasapogenol A could be converted into soyasapogenols B and E in the metabolic process. These results suggest that it is feasible to use soyasapogenols as functional ingredients in nutraceuticals or food formulations.

## 1. Introduction

Soyasaponins are oleanene-type triterpenoid saponins, and they can be mainly categorized into two groups, A and B, according to their respective aglycones, soyasapogenol A and soyasapogenol B [1]. Soybeans and soy foods are the major sources of dietary soyasaponins consumed by humans. An increasing number of investigations have demonstrated the health-promoting activities of dietary soyasaponins, such as lowering serum lipids [2,3] and inhibiting the growth of cancer cells [4]. However, to date, detailed information on the fate or bioavailability of soyasaponins in the guts of animals and humans is generally lacking. Previous studies have suggested that soyasaponins are poorly absorbed early in the GI tract and that they are mainly metabolized in the colon, which means that the microbiota is the main agent responsible for their hydrolysis and the production of sapogenols [5,6]. In this respect, it is important to note that diverse sapogenols have demonstrated superior bioavailability and bioactivity compared to their precursor saponin due to their more favorable chemical properties caused by a lack of sugar chains [7,8]. Soyasapogenols have been reported to possess a wide range of health benefits, including anticancer, antivirus, and antioxidant activities, as well as hepatoprotective and cardiovascular protective effects [3,9,10]. However, evidence demonstrating that soyasapogenols are systemically bioavailable in the conjugated form following microbial metabolism in the colon is still limited [8]. Therefore, it is essential to elucidate the metabolism and bioavailability of soyasapogenols in order to understand their bioactivity and to properly design functional food ingredients.

Examining the kinetic dynamics of metabolism is an important method to reveal the physiological benefits and bioavailability of biologically active ingredients, such as saponins, after consumption [11,12]. Biologically active substances in the body are absorbed into the blood and need to reach a specific plasma concentration to be effective [8,12]. Therefore, plasma concentration data are essential for determining the pharmacokinetic (distribution and elimination) and pharmacodynamic (equilibration and sensitivity) characteristics of bioactive components.

The LC–MS technique combines high separation ability with the high selectivity of high-resolution mass spectrometry and the ability to provide information (molecular weight and structural information) on complex samples, and it is widely used to control quality standards for natural phytochemicals and/or their intestinal metabolites [13,14]. The composition of natural herbal medicines as well as the changes in their chemical composition before and after processing have been rapidly analyzed using UPLC–Q-TOF–MS/MS, making it an effective tool for identifying active ingredients, especially those of saponins and their derivatives, with improved sensitivity and accuracy [15,16,17].

The aims of the present work are to assess the bioavailability of dietary soyasaponins in rats by directly using soyasapogenols as a dietary component and to establish the concentrations and forms of the main metabolites following the consumption of soyasapogenol-rich diets. The findings of this study may provide further insights into the direct use of soyasapogenols as functional ingredients in food formulations.

## 2. Results and Discussion

### 2.1. Optimization of the UPLC–Q-TOF–MS/MS Method

The choices of mobile-phase pairs for gradient elution and column separation were investigated. As a result, the acetonitrile–water pair was selected as the mobile-phase pair for the method. In addition, 0.1% formic acid was added to the mobile-phase pair to facilitate the protonation or deprotonation of the analytes for the mass spectrometric detection of the analytes in the positive ionization mode. It was also found that the Waters ACQUITY UPLC® BEH C18 column (2.1 mm i.d. × 100 mm, 1.7 μM) had greater separation efficiency and produced a larger signal-to-noise ratio and a better peak shape; therefore, it was adopted for the method. To optimize the MS/MS parameters for the compounds, both positive and negative ionization modes were applied to the analyses of soyasapogenol and its derived metabolite samples using QTOF–MS/MS, and comprehensive information about these components was obtained. The positive ion mode gave good ionization, and collision-induced dissociation energy was optimized in the range of 20–30 ev. For all the compounds, the protonated molecules [M + H]^+^ and their sodium adduct [M + Na]^+^ ions were selected based on their high abundance and stronger fragmentation regularity. To confirm the results and the structures of the metabolites, an ESI–QTOF–MS/MS analysis was performed. The peak results were identified based on their MS spectral data (accurate mass and fragmentation patterns), a comparison to standard compounds, and a search of public online databases (DNP, Reaxys, and SciFinder). The presence of a particular type of metabolite was proposed based on the occurrence of a characteristic molecular ion peak, and a further analysis of its fragmentation ions implied the structure of the metabolite.

At first, individual standard solutions of soyasapogenols A and B were directly injected. The recorded first-order mass spectra of the positive ions are displayed in Figure 1. The retention times of soyasapogenols A and B were 10.23 min and 10.72 min, respectively. The excimer ion peak [M + H]^+^ m/z of soyasapogenol A was 475.2774. The main fragment ions under the high-energy conditions were observed at m/z 185.1321, 283.2627, 333.2782, and 337.2521. The excimer ion peak [M + H]^+^ m/z of soyasapogenol B was 459.2299, and the main fragment ions were observed at m/z 163.1106, 283.2627, 401.2284, and 429.3716. 

### 2.2. Comparison of Plasma Soyasapogenol Concentrations in Rats after Soyasapogenol Intake

#### 2.2.1. Pharmacokinetics of Oral Administration

The pharmacokinetic parameters of soyasapogenols A and B in the plasma of the rats 0–24 h after oral administration are summarized in Table 1. As shown in Figure 2, following the oral administration of soyasapogenol A, there were rapid increases in the plasma concentrations to 29 ± 10 and 7.0 ± 6 μg/mL at the dosage levels of 10 mg/kg and 20 mg/kg by gavage at 1.0 h postadministration, respectively. The plasma concentrations reached the maxima (C_max_) of 8.59 mg /L at 2.0 h postadministration in the case of the 10 mg/kg dose and 28.99 mg /L at 1.0 h postadministration in the case of the 10 mg/kg dose, and then they decreased. The results also demonstrate that the area under the plasma or serum concentration–time curve from 0 to 24 h, AUC_(0-∞)_, and C_max_ were both linear and proportional to the oral dose. Likewise, after the oral administration of soyasapogenol B, it was also rapidly absorbed, with a T_max_ of 2 h. At different concentrations, the C_max_ values were 26.37 mg/L and 40.29 mg/L, and the half-life T_1/2z_ was very similar, which demonstrates the linear kinetics absorption process of soyasapogenol B in the rats following oral administration. In addition, the plasma elimination half-life appeared to be independent of the dose. As shown, the area under the curve AUC_(0-∞)_ gives a profile identical to that of soyasapogenol A in Figure 2 (left panel), and the results revealed that they were positively correlated (R = 0.63, *p* < 0.001).

Regarding the absorption and elimination of the orally administrated soyasapogenols, as indicated in Figure 2, in each dosage group, the kinetic profile revealed rapid absorption with a one-peak plasma concentration versus time course, followed by a different decreasing pattern consisting of a distribution phase and an elimination phase between the different dosage groups. This result is consistent with previous investigations [5], which demonstrated that the bioavailability of soyasapogenols is better than that of the corresponding soyasaponins and that the absorption of soyasapogenol B is better than that of soyasapogenol A. In this work, the results indicate that the time needed for the soyasapogenol A administered to the rats to reach peak concentration was 1 to 3 h, whereas the time needed for the soyasapogenol B administered to the rats to reach peak concentration was 8 h. 

We previously reported the in vivo absorption characteristic properties of soyasapogenols A and B at different locations in four intestinal segments of mice—the duodenum, jejunum, ileum, and colon. The results demonstrated that, in terms of the apparent absorption coefficient (P_app_) and the absorption rate constant (K_a_), the rank order of the absorption of soyasapogenol from various administration sites was as follows: jejunum > duodenum > ileum > colon, and the absorption capacity of soyasapogenol A exposure significantly and concentration-dependently accumulated over time compared to soyasapogenol B [18]. The results suggest that some transporters may be involved in the regulation of soyasapogenol B absorption in addition to concentration gradient-dependent diffusion.

#### 2.2.2. Pharmacokinetics of Intravenous Injection

Following the intravenous administration of a bolus dose of 3 mg/kg soyasapogenols to the rats, the elimination of soyasapogenol also followed a two-compartment or three-compartment model of elimination (Figure 3). 

The plasma concentration–time curve of the soyasapogenol intravenous injection is shown in Figure 3. The results indicate that the absorption of aglycone through the dorsal foot vein injection in the rats was very fast, within 0–2 h; the blood concentration decreased rapidly, and then the speed of its clearance slowed down until most of it had disappeared in vivo after 24 h. 

The main metabolic parameters of the soyasapogenols obtained after the intravenous administration are shown in Table 2. The soyasapogenols disappeared rapidly from the blood after the intravenous injections, reaching their peak at 0.03 h. The half-lives T_1/2z_ of soyasapogenol A were 3.54 h and 3.48 h, and the half-lives T_1/2z_ of soyasapogenol B were 5.14 h and 5.53 h, which indicates that the half-life of the soyasapogenols after the intravenous injections was fixed and independent of the concentration. The C_max_ and AUC_(0-∞)_ of the soyasapogenols were positively correlated with their concentrations. The metabolic regularity was similar to that of mangiferin aglycone [19], which undergoes extensive phase II metabolism and predominantly exists as glucuronidation and sulfation metabolites in vivo, and aglycone exhibits favorable pharmacokinetics properties, as partly indicated by its intermediate bioavailability.

### 2.3. Bioavailability of Soyasapogenols

The bioavailability (F) of bioactive components in vivo has been frequently used to investigate their absorption, metabolism, tissue distribution, and bioactivity [2,8]. The absolute oral bioavailability was calculated from the percentage ratio of the AUCs derived from the plasma soyasapogenol concentrations after oral and I.V. administration of the different sources of soyasapogenols to the rats. As a result, for soyasapogenol A, the F values were 73.10% and 67.34% at the dosages of 10 mg/kg and 20 mg/kg, respectively, while for soyasapogenol B, the F values were 60.94% and 69.01% at the dosages of 25 mg/kg and 50 mg/kg, respectively (Table 1). 

The high molecular weight and poor membrane permeability of saponins cause their low bioavailability, hence restricting their application as a drug candidate [20]. For example, the oral bioavailability of ginsenosides Ra3, Rb1, and Rd has been found to be 0.1–0.2%, whereas that of ginsenosides Re, Rg1, and notoginsenoside R1 has been found to be 0.2–0.6% [21]. An investigation has demonstrated that the bioavailability of soyasapogenols is better than that of the corresponding soyasaponins and that the bioavailability of group B soyasaponins is better than that of group A soyasaponins [5]. Previous research has also proved that sapogenols are present in human plasma following microbial metabolism in the colon; furthermore, soyasapogenol B has been found to be a major metabolite, and it has been found in the conjugated form [7]. It should be pointed out that a previous investigation has demonstrated that the amount (%) of soyasapogenol B that permeates into the basolateral side from the Caco-2 cell monolayer apical side is much higher than that of soyasapogenol A [5]. However, in the present study, the result revealed that the absorption of soyasapogenol B was not much higher than that of soyasapogenol A, and this is not in agreement with previous studies. This may be attributable to the different dosages used, as well as to the different experimental models; our data were obtained using in vivo experimental approaches, while the previously reported data were obtained using an in vitro Caco-2 cell line as a model system.

It was demonstrated that methylation, sulfate conjugation, S-cysteine conjugation, sulfate conjugation, and glycine conjugation were the main metabolic pathways in vivo for sulfate-related metabolites, indicating that the phase II reaction plays an important role in transforming soyasapogenols to less toxic metabolites, which has also been observed in other types of saponins [22,23,24].

As shown in Figure 1, the only structural difference between soyasapogenol A and soyasapogenol B is the presence of a hydroxyl group at position 21. Further research is needed to elucidate the mechanism by which this single hydroxyl group affects absorption. 

It can be seen that the absorption of the soyasapogenols in the rats after intragastric administration follows a bimodal pattern that is similar to that of other sapogenols of ginsenosides. It is well-known that various endogenous and exogenous compounds undergo enterohepatic circulation and intestinal metabolism, which determine their fate in an organism. The absorption of saponin components across the gastrointestinal wall membrane may be affected by multiple factors, such as solubility, interactions with other dietary ingredients, molecular transformations, different cellular transporters, metabolism, and interactions with the gut microbiota, resulting in changes in their bioavailability.

For dietary soyasaponins and soyasapogenols that are excreted extensively into bile, an insight into the magnitude of enterohepatic circulation is of crucial importance, as it significantly affects pharmacokinetic parameters, such as plasma half-life and the area under the plasma concentration–time curve (AUC), as well as the estimates of bioavailability. Therefore, the findings of this study provide further insights into the direct use of soyasapogenols as functional ingredients in food formulations. However, their poor solubility and undesirable taste merit more studies in order to resolve these problems.

### 2.4. Identification of Liver Metabolites of Soyasapogenols

#### 2.4.1. Metabolite Identification and Metabolic Characteristic Elucidation of Soyasapogenol A

UPLC–QTOF–MS was used to identify the liver metabolites of soyasapogenol A in the rats, and the extraction ion chromatograms are shown in Figure 4. According to the relative molecular weight value of each metabolite and the fragment ion information of the secondary mass spectrum, the possible structure of each metabolite was speculated according to the normal metabolic reaction in vivo. A total of 16 metabolites named M1–M16, principally glucuronides, were identified in the plasma of the rats after oral and intravenous administration. Their identities, summarized in Table 3, were determined using UPLC–QTOF–MS in the consecutive reaction monitoring mode.

#### 2.4.2. Metabolite Identification and Metabolic Characteristic Elucidation of Soyasapogenol B

Moreover, using the same analysis conditions as those used for the analysis of the soyasapogenol A metabolites, as shown in Table 4 and Figure 4, a total of 15 of soyasapogenol B metabolites, named M1–M15, were identified unambiguously by comparing their retention times, mass accuracies, and fragmentation behaviors with data from the corresponding reference standards, or they were tentatively characterized based on MS data and reference to the available data in the literature [5,7].

### 2.5. Metabolic Pathway of Soyasapogenols In Vivo

In this work, 16 compounds were detected for soyasapogenol A metabolites, and 15 compounds were observed in soyasapogenol B metabolite samples. Summarizing all of the results presented here, the proposed metabolic pathways of the soyasapogenols in vivo are illustrated in Figure 5. The major biotransformation of the soyasapogenols involved oxidation, dehydrogenation, hydrolysis, dehydration, and deoxygenation reactions, as well as other phase I reactions, in the metabolic process; furthermore, phosphorylation, sulfation, gluconaldehyde acidification, and conjugation with cysteine, as well as other phase II reactions, also occurred in the metabolic process. According to the reactions, M1–M6 are one-step metabolites; M7–M9 are two-step metabolites of M1 conjugated with cysteine, cysteine sulfhydryl, and acetylcysteine; M10–M11 are two-step metabolites of M2; M12–M14 are two-step metabolites of M4; and M15–M16 are two-step metabolites of M6. In addition, soyasapogenol A can be metabolized into soyasapogenol B and soyasapogenol E under certain conditions. 

## 3. Materials and Methods

### 3.1. Chemicals and Reagents

HPLC-grade methanol and analytical-grade ethanol were purchased from Shanghai Lingfeng Chemical Reagents Co., Ltd. (Shanghai, China). 

Soyasapogenol A and B standards (98 %+), β-glucuronidase, and sulfatase were purchased from Sigma-Aldrich (Shanghai) Trading Co., Ltd.

Pentobarbital sodium (AR, 98 %) was bought from Nanjing Jiancheng Biological Engineering Research Institute (Nanjing, China). Sodium citrate as an anticoagulant (4%) and Krebs–Ringer buffer were bought from Beijing Leagene Biotech Co., Ltd. (Beijing, China). Saline (0.9%) was obtained from Shandong Kangning Pharmaceutical Co., Ltd. (Liaocheng, China). Formononetin (FMN) was purchased from Yuanye (Shanghai, China). Food-grade sodium carboxymethyl cellulose (CMC) was purchased from Shanghai Aladdin Biochemical Technology Co., Ltd. (Shanghai, China).

All other reagents were of analytical grade, and ultra-pure water was used throughout the experimentation.

### 3.2. Preparation of Soyasapogenol Standard Solution

The stock solution of soyasapogenols A and B (IS) was prepared by dissolving 1.00 mg powder in 1.00 mL of methanol to a 1.00 mg/mL concentration. The working solution of IS was prepared via a 1/10 dilution of the stock solution in methanol to a concentration of 100 μg/mL, and it was stored in a refrigerator at 4 °C for later use.

### 3.3. Preparation and Application of Dosing Solutions

Soyasapogenol was prepared via the acid hydrolysis of the crude extract according to our previous methodology [25]. For the preparation of the soyasapogenol samples for oral and intravenous administration, 0.5% (*w/v*) sodium carboxymethyl cellulose (NaCMC) was dissolved in pure water as a vehicle; then, the soyasapogenols were added to the solution under continuous magnetic stirring to the desired concentration. For the intestine experiment, the in vivo intestinal perfusion consisted of soyasapogenol in Krebs–Ringer bicarbonate buffer (Beijing Leagene Biotech Co., Ltd., Beijing, China). Soyasapogenol was dissolved in 500 ml Krebs–Ringer bicarbonate buffer at two concentrations of 10 mg/L and 50 mg/L.

### 3.4. Animals

Twenty-four Sprague Dawley adult male rats (250 ± 20 g) were purchased from SPF Beijing Vital River Laboratory Animal Technology Co., Ltd. (Beijing, China). The eleven-week-old male Sprague Dawley rats were maintained in a ventilated room at 22 ± 3°C and 40–60% humidity with a 12 h light–dark cycle. The rats were fasted for 12 h and allowed to drink freely before the experiment. 

This study was approved by the Experimental Animal Ethics Committee of the Shanghai Jiao Tong University, and it strictly followed the instructions of the Organizational Guidelines and Ethics Guidelines of the Experimental Animal Ethics Committee (No. IACUC, approval no. A2019086).

### 3.5. Metabolic Kinetics and Bile Metabolism Experiment of Soyasapogenols

#### 3.5.1. Oral Administration

The experiments were performed according to Kamo et al [5]. The rats were divided into three groups (*n* = 8). All rats were fasted for 14–15 h prior to soyasapogenol A or B administration. All rats were orally administered soyasapogenol A or B via direct stomach intubation. The dosages of the test samples are summarized in Table 5.

#### 3.5.2. Intravenous Injection Experiment

The rats’ dorsal foot veins were cannulated with a standard syringe needle to administer the soyasapogenol. As presented in Table 5, the dosages of soyasapogenol A were 10 mg/kg and 20 mg/kg, and the dosages of soyasapogenol B were 25 mg/kg and 50 mg/kg. Blood samples were collected from the tail vein just prior to soyasapogenol administration (0 h) and 0.03, 0.08, 0.25, 0.5, 1, 2, 4, 6, 8, and 24 h after soyasapogenol loading; blood soyasapogenol levels were measured immediately. The collected blood sample volume was approximately 200 μL, and the samples were subjected to the same sample pretreatment procedure as described below.

#### 3.5.3. Perfusion of the Small Intestine

The preparation of the isolated, perfused small bowel was performed according to Guo et al. [19], with minor modifications, and the dosages of the test samples are presented in Table 5. Briefly, before the experiment, the rats were fasted for 12 h and supplied with distilled water. The rats were subjected to bile metabolism experiments, intraperitoneally injected with 1% sodium pentobarbital (40 mg/kg) as anesthesia, and placed on an electric heating pad after anesthesia administration to ensure a constant temperature of 37 °C; their backs were fixed to the operating table. The abdominal cavity was cut along the midline of the abdomen, with an incision of 3~4 cm, and the position of each intestinal segment was determined; the location of the bile was found, and a thin catheter was carefully inserted to collect the bile. Afterwards, the duodenum was incised at both ends, and the tubes were intubated and ligated with surgical sutures. The intestinal materials were slowly washed out with preheated normal saline at 37 °C. After the normal saline was drained, a constant flow pump was connected, and Krebs buffer was used first. The Ringer buffer (preheated at 37 °C in advance) was used to equilibrate each intestinal segment at a flow rate of 0.2 mL/min for 30 minutes, and then soyasapogenol perfusion fluid was used to start the experiment. Bile samples were collected in centrifuge tubes, and they were moistened during perfusion. A cotton cloth with physiological saline covered the abdominal cavity to keep it moist, and 1% sodium pentobarbital was given intermittently to maintain the anesthesia.

Blood samples were collected from the tail vein at 0 (pre-dose), 0.25, 0.5, 1, 2, 4, 6, 8, and 24 h after administration. The sample volume was 200 μL. The samples were immediately transferred to a centrifuge tube and treated with a sodium citrate anticoagulant. This fraction was then centrifuged at 10,000 rpm for 10 min, and the upper plasma was collected and placed in a refrigerator at −80 °C for later use.

### 3.6. Processing and Analysis of Plasma Samples

The soyasapogenol concentrations in the rat plasma solutions were determined according to the method published in [5]. At first, the plasma samples were thawed to 4 °C and transferred (20 to 100 μL) to an Eppendorf tube. After adding the same volume of 0.2 mol/L sodium acetate buffer to each sample, 3 µL aliquots of glucuronidase (Sigma; extracted from Helix pomatia, containing 95,600 units/mL glucuronidase and 1100 units/mL sulfatase) were added to analyze the glucuronyl- and sulfatide-conjugated forms of the soyasapogenols, and the mixture was incubated at 37 °C for 12 h. The samples were mixed thoroughly, and then they were centrifuged at 10,000 rpm for 15 min. The supernatant was blown with nitrogen gas until it was dry, and residual supernatant was dissolved with 1 mL methanol. Subsequently, after filtration, the samples were analyzed according to the method published in [5].

### 3.7. Processing and Analysis of Bile Samples

The bile samples were taken from the perfusion syringe before and after the perfusion experiment. Effluent perfusate samples were collected every 15 min for 120 min in tared 3 mL centrifuge tubes. Subsequently, 3 times the amount of acetonitrile was added to precipitate proteins. The supernatant was dried under nitrogen flow and re-dissolved in 400 μL acetonitrile with an ultrasonic bath for 5 min. Next, the samples were centrifuged for 10 min at 12,000× *g* to obtain the supernatant. The final step was to allow the samples to pass through a 0.22 μm millipore filter for an UPLC–QTOF–MS analysis. The processing of the blank group was the same as that of the experimental group. 

### 3.8. Instrumentation and LC–MS/MS Analytical Conditions

The UPLC–QTOF–MS/MS analysis was performed on an ACQUITY UPLC I-Class (Waters, Milford, MA, USA) system, which was coupled with tandem quadrupole (Xevo TQ-S) mass spectrometry (UPLC–Xevo TQ-S–MS/MS). 

Gradient UPLC elution was carried out on the column BECH C18 (100 mm × 2.1 mm, 1.7 μm, Waters, Milford, MA, USA) with deionized water containing 0.1% formic acid as mobile-phase A and acetonitrile (LC grade) containing 0.1% formic acid as mobile-phase B. The column temperature was kept at 45 °C, and the flow rate was set to 0.4 mL/min. The gradient elution program was finally optimized as follows: 5–20% B from 0 to 3 min, 20–100% B from 3 to 10 min, 100% B from 12 to 15 min, and 100–95% B from 15 to 19 min. The samples were kept at 20 °C in the auto-sampler manager, and the injection volume was 1.0 μL.

### 3.9. Mass Spectrometry Analysis Using Xevo G2-XS QTOF Instrument 

A mass spectrometry analysis of the soyasapogenol metabolites was completed using a Xevo G2-XS QToF (Waters Corporation, USA) instrument. All mass spectrometry analyses were performed in the positive ionization mode, with mass spectra acquired over the 50 to 1000 m/z range. The following parameter settings were used: a capillary voltage of 2 kV; a cone voltage of 40 V; a turbo spray temperature of 450 °C; and a collision energy (CE) of 6 eV. Nitrogen was used as the nebulizer and the auxiliary gas, the atomizing gas flow rate was set to 900 L/h, and the taper hole blowback gas was set to 50 L/h. 

### 3.10. Data Processing

Curve-fitting simulations were performed using MicroCal Origin software, version 6.1 (MicroCal Software Inc., Northampton, MA, USA).

The metabolites of the soyasapogenols in the rat blood and bile samples were identified using the MetaboLynx™ (v4.1) program (Waters Corp., Milford, MA, USA) based on the UPLC–QTOF–MS chromatograms. In post-acquisition analyses, the MetaboLynx™ (v4.1) program could employ an extensive list of potential biotransformation reactions (e.g., oxidation, phosphorylation, sulfation, and conjugation) of the prototype molecule, in combination with the elemental compositions, to generate a series of extracted ion chromatograms (XICs). These mass spectral peaks, using a mass accuracy of two decimal places, were then compared between the control and experimental samples to eliminate the chromatographic peaks in the samples that also appeared in the blank matrix. Eventually, the metabolites that were derived from the major compounds in vitro and the absorbed prototype constituents were predicted using MetaboLynx software (Waters). The molecular formulas and structures of the metabolites investigated were elucidated based on the information of their precursor ions, fragmentation ions, and chromatographic retention times.

The bioavailability parameters were calculated by comparing AUC_oral_ postadministration with AUC_iv_ post-soyasapogenol I.V. injection:%Bioavailability = [(AUC_oral_) × Dose_iv_/(AUC_iv_) × Dose_oral_] × 100

The calculation formula is as follows: F = oral administration AUC_(0-∞)_/intravenous injection AUC_(0-∞)_.

Data are presented as the percentage change in LV thrombus minus cavity gray scale (%). Data were analyzed by using SigmaStat version 2.03 statistical software (SPSS Inc). The results are expressed as mean ± SEM. Differences between groups were analyzed using the one-way ANOVA Student–Newman–Keuls method; a value of *p* < 0.05 was considered statistically significant.

## 4. Conclusions

In conclusion, the present research provides useful information on the in vivo pharmacokinetics and bioavailability of soyasapogenols in rats, helps in the design of new nutraceuticals by providing kinetics-based data, and offers a better understanding of the fate and intermediary metabolism of dietary soyasapogenols. Further investigations are needed to identify the conditions that can enhance their solubility and to clarify their physiologic activities in the digestive organs.

## Figures and Tables

**Figure 1 molecules-28-00284-f001:**
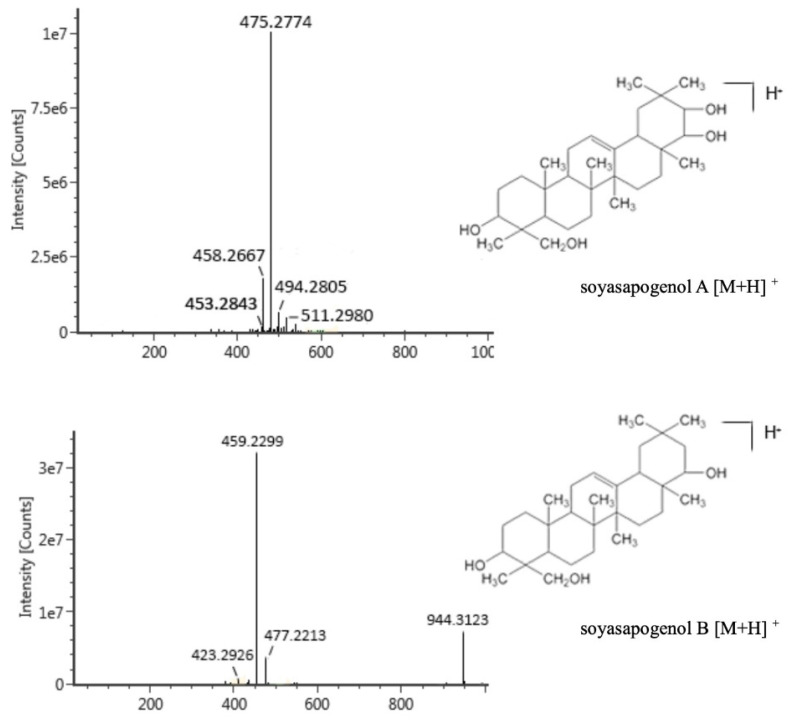
ESI–QTOF–MS/MS spectra of soyasapogenols A and B in the positive ion mode.

**Figure 2 molecules-28-00284-f002:**
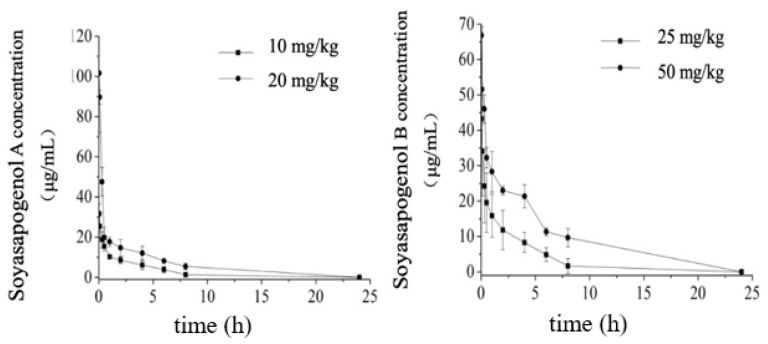
The concentration–time curve of soyasapogenol after intravenous administration in rat plasma (*n* = 6).

**Figure 3 molecules-28-00284-f003:**
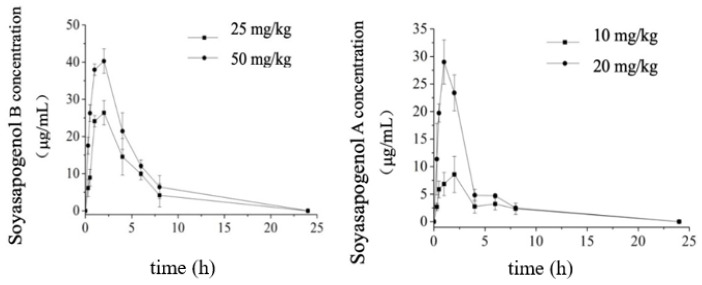
Plasma concentration–time curve of soyasapogenol after different dosages of oral administration. Left panel, soyasapogenol A: 10 and 20 mg/kg body weight (BW). Right panel, soyasapogenol B: 25 and 50 mg/kg BW. Values are mean ± SE; *n* = 6.

**Figure 4 molecules-28-00284-f004:**
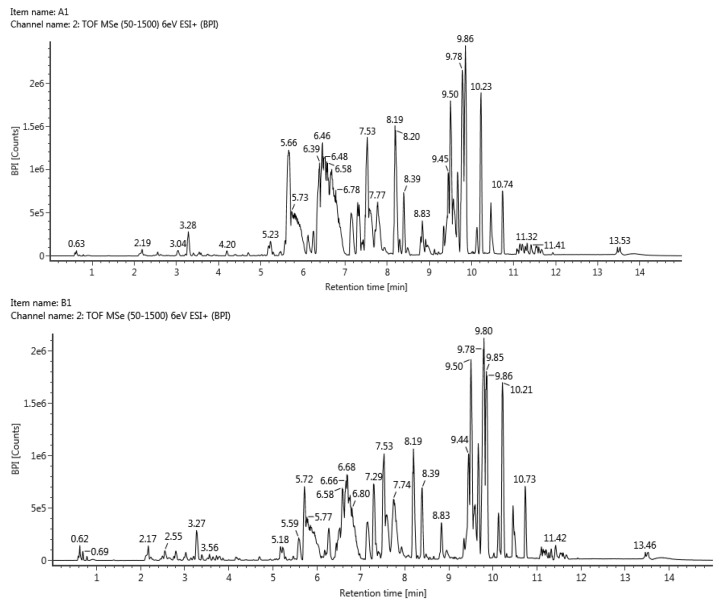
Extracted ion chromatograms of soyasapogenol metabolites in bile.

**Figure 5 molecules-28-00284-f005:**
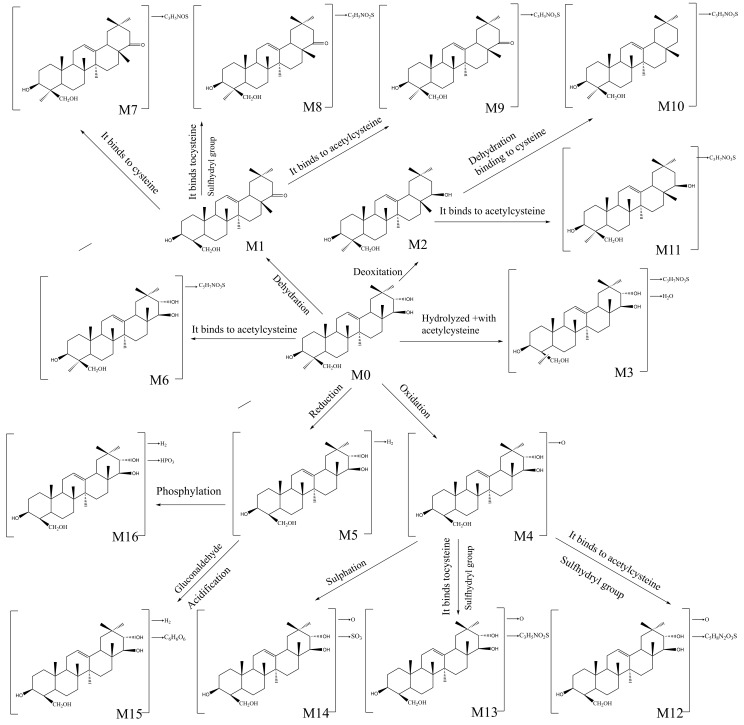
Proposed metabolism of soyasapogenols following oral and intravenous administration in rats.

**Table 1 molecules-28-00284-t001:** Summary of pharmacokinetic parameter values (mean, *n* = 6) for soyasapogenols A and B after oral (P.O.) dose administration.

Parameter	Unit	Soyasapogenol A		Soyasapogenol B	
		10 mg/kg	20 mg/kg	25 mg/kg	50 mg/kg
AUC_(0-t)_	mg/L*h	35.198	88.619	96.029	157.218
AUC_(0-∞)_	mg/L*h	50.700	95.507	110.513	178.341
T_1/2z_	h	2.552	1.966	2.288	2.266
T_max_	h	2	1	2	2
CL_z_/F	L/h/kg	0.394	0.209	0.181	0.112
C_max_	mg/L	8.59	28.99	26.37	40.29
Ratio of B/A in test sample		-		2.5	2.5
Ratio of B/A in plasma level (AUC)		-		2.17	1.87
Bioavailability (F,%)		73.10	67.34	60.94	69.01

C_max_: peak concentration; T_max_: peak time; T_1/2z_: biological half-life; AUC: area under concentration–time curve; CLz/F: clearance; Ratio of B/A: ratio of group B soyasapogenol B to soyasapogenol A in test samples or in plasma at the same concentration.

**Table 2 molecules-28-00284-t002:** Summary of pharmacokinetic parameter values (mean, *n* = 6) for soyasapogenols A and B after intravenous (I.V.) dose administration.

Parameter	Unit	Soyasapogenol A		Soyasapogenol B	
		10 mg/kg	20 mg/kg	25 mg/kg	50 mg/kg
AUC_(0-t)_	mg/L*h	56.029	114.473	76.214	154.392
AUC_(0-∞)_	mg/L*h	69.355	141.82	181.347	258.42
T_1/2z_	h	3.535	3.475	5.138	5.528
T_max_	h	0.03	0.03	0.03	0.03
CL_z_/F	L/h/kg	0.288	0.141	0.246	0.077
C_max_	mg/L	31.7	101.65	43.26	66.85

**Table 3 molecules-28-00284-t003:** Summary of the Mass Spectral Data of Soyasapogenol A and Its Metabolites Detected in Rat Bile.

The Serial Number	Metabolites	Molecular Formula	[M + H]^+^ m/z	Retention Time (min)	Fragment Ion Information	Metabolic Pathways
M0	Soyasapogenol A	C_30_H_50_O_4_	475.3803	10.28	185.1321, 283.2627, 333.2782, 337.2521	Prototype
M2	Soyasapogenol A–H_2_O	C_30_H_48_O_3_	457.3678	9.92	166.2149, 187.1435, 403.2801	Dehydration
M3	Soyasapogenol A–O	C_30_H_50_O_3_	459.3831	10.36	190.2349, 320.2433, 408.2959	Deoxidation
M4	Soyasapogenol A + H_2_O + C_5_H_7_NO_3_S	C_35_H_59_NO_8_S	654.4005	9.24	117.0907, 292.1592, 462.2684, 534.3287	Hydrolysis + with acetylcysteine
M5	Soyasapogenol A + O	C_30_H_50_O_5_	491.3742	9.73	129.1607, 255.3044, 273.2602, 347.3641	Oxidation
M6	Soyasapogenol A + H_2_	C_30_H_52_O_4_	477.3931	11.04	273.1810, 315.2651, 359.2912	Reduction
M7	Soyasapogenol A + C_5_H_7_NO_3_S	C_35_H_57_NO_7_S	636.3946	7.29	193.1584, 253.1942, 460.2505, 478.2607	Binding to acetylcysteine
	Soyasapogenol A–H_2_O + C_3_H_5_NOS	C_33_H_53_NO_4_S	560.3784	9.34	375.3043, 393.3139, 439.2520	Dehydration + binding to cysteine
M8	Soyasapogenol A–H_2_O + C_3_H_5_NO_2_S	C_33_H_53_NO_5_S	576.3732	7.63	119.0851, 285.2203, 306.1489, 522.2855	Dehydration + with cysteine sulfhydryl group
M9	Soyasapogenol A–H_2_O + C_5_H_7_NO_3_S	C_35_H_55_NO_6_S	618.3851	7.24	255.2099, 357.2779, 423.3239, 462.2661	Dehydration + binding with acetylcysteine
M10	Soyasapogenol A–O–H_2_O + C_3_H_5_NOS	C_33_H_53_NO_3_S	544.3844	9.73	273.2208, 485.2630	Deoxidation + dehydration + binding to cysteine
M11	Soyasapogenol A–O + C_5_H_7_NO_3_S	C_35_H_57_NO_6_S	620.4007	8.42	141.1271, 340.1576, 365.2829	Deoxidation + binding to acetylcysteine
M12	Soyasapogenol A + O + C_5_H_8_N_2_O_3_S	C_35_H_58_N_2_O_8_S	667.3980	9.92	337.2521, 459.2476, 565.3347, 609.3608	Oxidation + binding to cysteinylglycine mercapto
M13	Soyasapogenol A + O + C_3_H_5_NO_2_S	C_33_H_55_NO_7_S	610.3754	9.26	89.0595, 133.0857, 247.2050, 333.2428	Oxidation + binding to cysteine sulfhydryl groups
M14	Soyasapogenol A + O + SO_3_	C_30_H_50_O_8_S	571.3298	8.73	209.1163, 277.2159, 335.2600, 427.3197	Oxidation + sulfation
M15	Soyasapogenol A + H_2_ + C_6_H_8_O_6_	C_36_H_60_O_10_	653.4253	9.68	449.2132, 491.2973, 535.3234, 579.3498	Reduction + gluconaldehyde acidification
M16	Soyasapogenol A + H_2_ + HPO_3_	C_30_H_53_O_7_P	557.3606	9.20	124.9994, 291.2311, 339.1717	Reduction + phosphorylation

**Table 4 molecules-28-00284-t004:** Summary of the Mass Spectral Data of Soyasapogenol B and Its Metabolites Detected in Rat Bile.

The Serial Number	Metabolites	Molecular Formula	[M + H]^+^ m/z	Retention Time (min)	Fragment Ion Information	Metabolic Pathways
M0	Soyasapogenol B	C_30_H_50_O_3_	459.3838	10.94	163.1106, 283.2627, 401.2284, 429.3716	Prototype
M1	Soyasapogenol B + H_2_O	C_30_H_52_O_4_	477.3926	11.04	185.1252, 342.2854, 370.3192	Hydrolysis
M2	Soyasapogenol B–H_2_O + C_3_H_5_NOS	C_33_H_53_NO_3_S	544.3848	9.81	199.1472, 239.2363, 515.3103	Dehydration + binding to cysteine
M3	Soyasapogenol B + H_2_O + HPO_3_	C_30_H_53_O_7_P	557.3601	9.19	139.1112, 253.1939, 413.3035, 507.2914	Hydrolysis + phosphorylation
M4-1	Soyasapogenol B–H_2_ + C_3_H_5_NOS	C_33_H_53_NO_4_S	560.3759	9.06	89.0596, 149.0956, 411.2872, 465.3320	Dehydrogenation + binding to cysteine
M4-2	Soyasapogenol B–H_2_O + C_3_H_5_NO_2_S	C_33_H_53_NO_4_S	560.3778	9.34	145.1008, 199.1474, 427.3192, 490.2986	Dehydration + with cysteine sulfhydryl group
M5-1	Soyasapogenol B + O-H_2_ + C_3_H_5_NOS	C_33_H_53_NO_5_S	576.3734	7.84	133.0857, 273.2206, 481.2611, 525.2875	Oxidation + dehydrogenation + binding to cysteine
M5-2	Soyasapogenol B–H_2_ + C_3_H_5_NO_2_S	C_33_H_53_NO_5_S	576.3734	8.41	215.1785, 339.2672, 409.3090, 488.2865	Dehydrogenation + binding to the thiol group of cysteine
M6-1	Soyasapogenol B–H_2_ + O_2_ + C_3_H_5_NOS	C_33_H_53_NO_6_S	592.3690	6.21	121.0644, 179.1064, 351.2526, 427.2812	Dehydrogenation + deoxidation + binding to cysteine
M6-2	Soyasapogenol B + O–H_2_ + C_3_H_5_NO_2_S	C_33_H_53_NO_6_S	592.3692	8.29	152.0374, 337.2518, 462.2663, 480.2770	Oxidation + dehydrogenation + binding to cysteine sulfhydryl groups
M7	Soyasapogenol B–O + C_5_H_7_NO_3_S	C_35_H_57_NO_5_S	604.4046	9.33	149.0956, 231.1738, 275.1996, 526.2961	Deoxidation + binding to acetylcysteine
M8	Soyasapogenol B–H_2_ + C_5_H_7_NO_3_S	C_35_H_59_NO_5_S	606.4217	7.76	103.0749, 276.1252, 339.2673, 464.2819	Deoxidation + reduction + binding to acetylcysteine
M9	Soyasapogenol B–O + H_2_ + C_5_H_7_NO_3_S	C_35_H_55_NO_6_S	618.3843	7.23	163.1113, 243.2094, 393.2793, 502.2581	Dehydrogenation + combining with cysteinylglycine mercapto
M10	Soyasapogenol B + C_5_H_7_NO_3_S	C_35_H_57_NO_6_S	620.3997	7.66	261.1839, 357.2776	Binding to acetylcysteine
M11	Soyasapogenol B–H_2_ + C_5_H_8_N_2_O_3_S	C_35_H_56_N_2_O_6_S	633.3940	9.04	89.0594, 287.1460, 407.2023, 520.3314	Dehydrogenation + binding to acetylcysteine
M12	Soyasapogenol B + O–H_2_ + C_5_H_7_NO_3_S	C_35_H_55_NO_7_S	634.3794	6.43	215.1424, 355.2622, 480.2767, 503.3029	Oxidation + dehydrogenation + gluconaldehyde acidification
M13	Soyasapogenol B + O + C_5_H_7_NO_3_S	C_35_H_57_NO_7_S	636.3899	7.29	133.0853, 210.0425, 289.2152, 462.2657	Oxidation + gluconaldehyde acidification
M14	Soyasapogenol B–H_2_ + O_2_ + C_5_H_7_NO_3_S	C_35_H_55_NO_8_S	650.3738	6.86	97.0643, 205.1216, 353.2458, 522.2850	Dehydrogenation + deoxidation + gluconaldehyde acidification
M15	Soyasapogenol B + H_2_O + C_6_H_8_O_6_	C_36_H_60_O_10_	653.4280	9.68	221.2259, 361.1606, 518.3208, 546.3546	Hydrolysis + gluconaldehyde acidification

**Table 5 molecules-28-00284-t005:** Dose and composition of the tested samples in each treatment group.

Treatments	Bulk Solution	Soyasapogenol Composition	Low Dosage		High Dosage	
			Dose (μ mol/kg BW)	Dose (mg/kg BW)	Dose (μ mol/kg BW)	Dose (mg/kg BW)
Oral administration (P.O.)	0.5% (*w*/*v*) sodium carboxymethyl cellulose solution used as a vehicle	A	21	10	42	20
		B	55	25	109	50
Intravenous injection (I.V.)	0.5% (*w*/*v*) sodium carboxymethyl cellulose solution used as a vehicle	A	21	10	42	20
		B	55	25	109	50
Perfusion of the small intestine	Krebs–Ringer bicarbonate buffer	A(B)	10 mg/L(50 mg/L)			

BW, body weight.

## Data Availability

Not applicable.

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
