# Peer review of "The Fate and Intermediary Metabolism of Soyasapogenol in the Rat"

_molecules, 2022, doi:10.3390/molecules28010284_

Round 1
Reviewer 1 Report
The manuscript is an interesting approach for the determination of the bioavailability of dietary soyasaponins in rats by evaluating the concentrations and forms of the main metabolites following the consumption of soyasapogenol rich diets.
Just two comments
A graphical abstract or a schematic representation of the proposed method should be added to the manuscript to help readers approaching the contents
The introduction needs to be revised. References about instrumentations and techniques applied in this field must be added.
Reviewer 2 Report
1. More information about soyasapongenol is required to be added in the introduction with more cited literature, which is missing in the introduction part and also in the result section.
2. What are the potential follow-ups for utilizing this component for better absorptions?
3. Authors are advised to clearly mention the biochemical environment of the site of application.
4. It is required to clearly mention the prerequisites for the breakdown of the parent compounds and the role of other interfering metabolites/enzymes present in various sites of application.
5. Specify the involvement of any gut microflora in the breakdown of soyasapongenol in a detail section .
Author Response
Please check the attacked file for details.
